# Influence of Rotor Dynamic Scattering on Helicopter Radar Cross-Section

**DOI:** 10.3390/s20072097

**Published:** 2020-04-08

**Authors:** Zeyang Zhou, Jun Huang

**Affiliations:** School of Aeronautic Science and Engineering, Beijing University of Aeronautics and Astronautics, Beijing 100191, China; junh@china.com

**Keywords:** helicopter, rotor, radar stealth, radar cross-section, dynamic electromagnetic scattering

## Abstract

With the continuous improvement and development of armed helicopters, the research on their stealth characteristics has become more and more in depth. In order to obtain the complex effect of stealth characteristics caused by the high-speed rotation of rotor-like components, a dynamic scattering method (DSM) is presented. Rotation speed, azimuth, elevation angle, pitching angle, and rolling angle are studied and discussed in detail. The results show that the electromagnetic scattering characteristics of the main rotor and tail rotor are dynamic and periodic. This period characteristic is related to the rotation speed and attitude angle of the rotor. The radar cross-section (RCS) of the helicopter varies greatly at different observation angles and attitude angles, but the dynamic electromagnetic scattering effect caused by the main rotor and tail rotor cannot be ignored. The presented DSM is effective and efficient for studying the dynamic RCS of the rotor-type parts of a helicopter or the whole machine.

## 1. Introduction

Modern helicopters are moving toward high speed and stealth while pursuing more excellent aerodynamic and handling characteristics [1,2], such as X2 (Sikorsky, City of Stateford, CT, USA), Tiger (Eurocopter, Paris, France), and RAH-66 (Boing, Chicago, USA) [3]. Due to the rotating parts such as the main rotor and the tail rotor, the research on the dynamic electromagnetic scattering characteristics of helicopters has always been a difficult point.

Low detectability technology is widely used in helicopters, fighter jets, and missiles, requiring the consideration of radar stealth for important components from the beginning, including cockpits, exhaust systems, and rotors [4,5]. The main rotor is the most important aerodynamic part of the helicopter, while the high-speed rotating rotor makes the aircraft have unique flight behavior and maneuverability [6,7]. However, the aerodynamic design of the rotor also affects its stealth characteristics [8,9]. Using the Maxwell equations as the main control equations, a set of numerical calculation methods for the rotor radar cross-section (RCS) characteristics based on the time-domain finite volume method is established. For the calculation of static electromagnetic scattering characteristics, the RCS solution of the helicopter is the same as the RCS solution of the fighter or unmanned fighter [10,11]. However, in the face of the high-speed rotation of the rotor, the static calculation obviously cannot reflect or meet the actual dynamic scattering characteristics of the helicopter.

Using the quasi-static principle (QSP) to discrete time series, the periodic rotary motion of the blades is decomposed into transient states, and the dynamic RCS response characteristics of the rotor are studied [12]. This method is simple and easy to understand and can be widely used in the motion simulation of rotor parts of helicopters or rotorcrafts [13,14], while it cannot achieve instantaneous calculations at a large number of points in time, nor can it reflect the continuity of rotor RCS changes. The geometric model of the rotor is established with NACA0012 as the blade airfoil, while the physical optics (PO) and an equivalent current method are used to calculate the rotor RCS under various motion conditions [15,16]. For less computer time, the grid transformation process using the body coordinate system and grid adaptive technology is applied to existing cavitation algorithms [17]. For multi-section topographical surfaces, a narrow surface element method is used to update a large amount of mesh data to optimize its radar cross-section [18]. Grid transformation technology can also be found in global ocean models and constant volume transformations [19,20]. Therefore, when considering the rotary motion of the rotor [21,22], the ideas of these mesh regeneration techniques can be implanted into a new dynamic simulation method and then used to calculate the dynamic RCS of the rotor.

Historically, research on the radar stealth of helicopters and rotor components is mostly based on quasi-static methods, and their calculation results are also static RCS or a small number of quasi-static results [23,24]. The core of QSP is to discretize many sample states. This approach is very cumbersome and laborious when facing the paddle angle, fuselage attitude angle, and multi-rotor. The relativistic law converts the rotation angle to the observation angle, which cannot deal with the situation when the fuselage or multiple rotors exist at the same time [15]. To overcome the shortcomings of these traditional methods, considering the complex effects of rotor rotation and tilt design [25,26], this paper attempts to establish a method for studying the dynamic electromagnetic scattering characteristics of the separate rotor and rotor plus its fixing device, and then it studies the radar stealth characteristics of helicopters and rotors at different attitude angles. It can be seen that studying the dynamic scattering of the rotor is of practical significance to the radar cross-section of the helicopter as a whole.

In this manuscript, the research method of rotor dynamic electromagnetic scattering is presented in Section 2. The models of the main rotor, tail rotor, and helicopter are built in Section 3. The static and dynamic RCS calculation results of the rotor and helicopter are given and discussed as shown in Section 4. Finally, this article is summarized as shown in Section 5.

## 2. Dynamic Scattering Method

The effect of the rotor on the radar stealth characteristics of the helicopter is shown in Figure 1, where *A*_b,r1_ is the angle between the adjacent blades of the main rotor, *A*_r,r1_ is the rotation angle of the main rotor, *t* is time, and subscript 0 is the initial time. The main rotor is designed with 4 blades, while the tail rotor has 3 blades. In the initial state, the main rotor disc is parallel to the horizontal plane, while the tail rotor disc is perpendicular to the horizontal plane.

### 2.1. Dynamic Electromagnetic Scattering

The flow chart of the entire method is shown in Figure 2. When importing different models (including rotors and helicopters), it is necessary to analyze the grid information and then set the azimuth and elevation angles and start the time cycle. Determine the position vector transformation according to the rotation of the rotor component, then update the mesh and extract the bin and edge information to perform RCS calculation. Integrate the calculation results and then judge whether each cycle is over, and output the dynamic RCS.

At the initial moment, the helicopter model here consists of the main rotor, tail rotor, and fuselage, which can be described as follows:(1)$$m_{h}\left(t=0\right)=\left\{m_{r1}\left(t=0\right),m_{r2}\left(t=0\right),m_{\mathrm{fus}}\left(t=0\right)\right\}$$
where *m*_r1_ is the model of the main rotor, *m*_r2_ is the model of the tail rotor, *m*_fus_ is the model of the fuselage. 

In the PO method, the electromagnetic wave is irradiated on the scatterer, and an induced current is generated in the illuminated area. According to the Stratton–Chu integral equation, the scattered electric field in the far field can be expressed as:(2)$$E_{s}^{\left(m_{h}(t=0)\right)}=\frac{j\omega \mu_{0}}{4\pi }\frac{\exp\left(-jkR\right)}{R}\int_{s^{\prime }}s\times \left(M\left(r\right)+Z_{0}s\times J\left(r\right)\right)\exp\left(jks\cdot r\right)ds^{\prime }$$
where *ω* represents the electromagnetic wave angular frequency, *μ*_0_ is the permeability coefficient, *k* refers to the wave number of free space, *R* is the distance from the observation point to the origin, *Z*_0_ represents the intrinsic impedance, ***s*** is the unit vector of the scattered wave, ***r*** represents the position vector of the observation point, *M*(***r***) is the equivalent magnetic surface current of the panel of the illuminated area, and *J*(***r***) is the equivalent electric surface current of the panel of the illuminated area:(3)$$\left\{\begin{array}{l} M\left(r\right)=n\times E\left(r\right)\\ J\left(r\right)=n\times H\left(r\right) \end{array}\right.$$

Discretize the target surface space using triangular facets:(4)$$M_{f,h}\left(t=0\right)=\left\{M_{f,r1}\left(t=0\right),M_{f,r2}\left(t=0\right),M_{f,\mathrm{fus}}\left(t=0\right)\right\}$$
(5)$$s_{I}⇐M_{f,h}\left(t=0\right)$$
where *s*_I_ is the illuminated area. When the rotor begins to rotate, the lighting area on its surface changes as shown in Figure 3, where *F*(*t*) is the facet at time *t*, which is composed of three vertices (*P*_1_,*P*_2_,*P*_3_).

For any vertex (*P_i_*) on any surface element of the main rotor surface, its coordinate information will continuously change as the rotor moves when treating the rotor as a rigid body:(6)$$\forall P_{i}={\left[x\left(P_{i}\right),y\left(P_{i}\right),z\left(P_{i}\right)\right]}^{T}\in F_{r1}\left|i\right.=1,2,\ldots ,N_{f,r1}$$
(7)$$\forall F_{r1}\in M_{f,r1}\left(t=0\right)$$
(8)$$M_{f}\left(F_{r1}\left(t\right)\right)=\left[\begin{array}{ccc} \cos A_{r,r1}\left(t\right) & -\sin A_{r,r1}\left(t\right) & 0\\ \sin A_{r,r1}\left(t\right) & \cos A_{r,r1}\left(t\right) & 0\\ 0 & 0 & 1 \end{array}\right]\cdot M_{f}\left(F_{r1}\left(t=0\right)\right)\left|\begin{array}{l} \gamma =\theta =0 \end{array}\right.$$
where *N*_f,r1_ is the number of facets of the main rotor. The treatment of the tilt of the main rotor disc plane is consistent with the attitude change of the helicopter; thus, the change of the main rotor disc plane is ignored here.

For the tail rotor, there are:(9)$$\forall P_{i}={\left[x\left(P_{i}\right),y\left(P_{i}\right),z\left(P_{i}\right)\right]}^{T}\in F_{r2}\left|i\right.=1,2,\ldots ,N_{f,r2}$$
(10)$$\forall F_{r2}\in M_{f,r2}\left(t=0\right)$$
(11)$$M_{f}\left(F_{r2}\left(t\right)\right)=\left[\begin{array}{ccc} \cos A_{r,r2}\left(t\right) & 0 & -\sin A_{r,r2}\left(t\right)\\ 0 & 1 & 0\\ \sin A_{r,r2}\left(t\right) & 0 & \cos A_{r,r2}\left(t\right) \end{array}\right]\cdot M_{f}^{xz}\left(F_{r2}\left(t=0\right)\right)\left|\begin{array}{l} \gamma =\theta =0 \end{array}\right.$$
(12)$$M_{f}^{xz}\left(F_{r2}\left(t=0\right)\right)=M_{f}^{z}\left(x\left(m_{r2}\left(t=0\right)\right)+X_{r2}\right)\left|\gamma =\theta =0\right.$$
(13)$$M_{f}^{z}\left(F_{r2}\left(t=0\right)\right)=M_{f}\left(z\left(m_{r2}\left(t=0\right)\right)-Z_{r2}\right)\left|\gamma =\theta =0\right.$$

When the pitching and rolling angles are both equal to 0, the fuselage model is considered unchanged:(14)$$M_{f}\left(F_{\mathrm{fus}}\left(t\right)\right)=M_{f}^{x}\left(F_{\mathrm{fus}}\left(t=0\right)\right)\left|\gamma =\theta =0\right.$$

After translating the rotated tail rotor to the initial position, the grid matrix of this helicopter can be obtained:(15)$$M_{f,h}\left(t\right)=\left\{M_{f,r1}\left(t\right),M_{f,r2}\left(t\right),M_{f,\mathrm{fus}}\left(t\right)\right\}\left|\gamma =\theta =0\right.$$
(16)$$s_{I}⇐M_{f,h}\left(t\right)$$

When the helicopter’s pitching angle starts to change, its grid matrix can be updated to:(17)$$M_{f}\left(m_{h}\left(t,\theta \right)\right)=\left[\begin{array}{ccc} \cos\theta & 0 & -\sin\theta \\ 0 & 1 & 0\\ \sin\theta & 0 & \cos\theta \end{array}\right]\cdot M_{f}\left(m_{h}\left(t\right)\right)\left|\begin{array}{l} \gamma =0 \end{array}\right.$$

When the rolling angle changes, the helicopter’s grid matrix can be described as:(18)$$M_{f}\left(m_{h}\left(t,\theta ,\gamma \right)\right)=\left[\begin{array}{ccc} 1 & 0 & 0\\ 0 & \cos\gamma & -\sin\gamma \\ 0 & \sin\gamma & \cos\gamma \end{array}\right]\cdot M_{f}\left(m_{h}\left(t,\theta \right)\right)$$

At this time, the illuminated area on the helicopter surface can be updated to:(19)$$s_{I}\left(t\right)⇐M_{f,h}\left(t\right)$$

Summing the electric fields of the triangle bins in the illuminated area, we have:(20)$$E_{s}{}^{\left(M_{f,h}(t)\right)}=\frac{jk}{4\pi }\frac{e^{-jkr}}{r}\sum_{p=1}^{N_{f,h}}\left(M_{s}^{p}+Z_{0}s\times J_{s}{}^{p}\right)\cdot I^{p}$$
where *N*_f,h_ is the number of triangular facets of the illumination area and ***I****^p^* is the feature integration of the *p*-th triangle face element, thus:(21)$$I=\int_{s}\exp\left[jkr\cdot \left(s-i\right)\right]ds^{\prime }$$
where ***i*** is the unit vector of the incident wave. The Gardon method [27] is used to describe the feature integration:(22)$$I^{p}=\frac{1}{jk\left|n\times w\right|\Delta A}\sum_{m=1}^{3}\left(n\times w\right)\cdot a_{m}\exp\left(jkr_{m}\cdot w\right)\sin c\left(\frac{1}{2}ka_{m}\cdot w\right)$$
where ***n*** refers to the outward unit normal vector of the surface, ***a**_m_* represents the *m*-th edge vector of the bin, and ***r****_m_* represents the mid-point position vector of the *m*-th edge. Noting that:(23)$$\left\{\begin{array}{l} w=s-i\\ \sin c\left(x\right)=\sin x/x \end{array}\right.$$

Further simplification of the feature integration:(24)$$I^{p}=\Delta A^{p}\exp\left(jkr_{0}\cdot w\right),\;\left|n\times w\right|=0$$
where *r*_0_ is the position vector of any point on the triangular facet, and Δ*A* represents the area of the triangular facet.

The physical theory of diffraction (PTD) is used to solve the edge diffraction of the target model [3,5]; then, the target total RCS is the RCS sum of all facets and all splits:(25)$$\sigma \left(t\right)={\left|\sum_{i=1}^{N_{F}(t)}\left(\sqrt{\sigma_{F}(t)}\right)\begin{array}{c} i \end{array}+\sum_{j=1}^{N_{E}(t)}\left(\sqrt{\sigma_{E}(t)}\right)\begin{array}{c} j \end{array}\right|}^{2},\;t\in \left[0,T_{\mathrm{ob}}\right]$$
where *σ* is the radar cross-section, subscript F represents the facet contribution, and E represents the edge contribution. *N*_F_ is the number of facets, *N*_E_ is the number of edges, and *T*_ob_ is the observation time boundary.
(26)$$T_{\mathrm{ob}}\geq \max\left\{t_{b,r1},t_{b,r2}\right\}\left|\right.\gamma =\theta =0$$
(27)$$t_{b}=\frac{A_{b}}{\omega_{r}}\cdot \frac{\pi }{180}$$
(28)$$A_{b}=\frac{360}{N_{b}}$$
where *t*_b_ is equal to the time it takes for the blade to rotate through the angle between two adjacent blades. When the attitude angle of the helicopter is not equal to 0, there are:(29)$$T_{\mathrm{ob}}\geq \max\left\{t_{b,\max},N_{b,r1}\cdot t_{b,r1},N_{b,r2}\cdot t_{b,r2}\right\}\left|\gamma \neq 0\begin{array}{cc} \begin{array}{cc} & \mathrm{or} \end{array} & \end{array}\theta \neq 0\right.$$
(30)$$t_{b,\max}=\max\left\{t_{b,r1},t_{b,r2}\right\}$$
(31)$$t_{w}=D/c$$
(32)$$\varepsilon_{0}=\min\left\{\left|t_{b,r1}-t_{w}\right|,\left|t_{b,r2}-t_{w}\right|\right\}$$
where *t*_w_ is the time from emitting electromagnetic waves to electromagnetic waves reaching the target surface, and *ε*_0_ is a custom time difference. The determination of *T*_ob_ is mainly to explore whether the electromagnetic scattering characteristics of the rotor are periodic.

### 2.2. Method Validation

The RCS calculations involved here are instantaneous and dynamic. For instantaneous or static RCS calculations, DSM is still based on PO + PTD, where the calculation results here are verified by PO + MOM (method of moment)/MLFMM (multi-level fast multipole method) in FEldberechnung bei Korpern mit beliebiger Oberflache (FEKO) simulation software on the tail rotor model, as shown in Figure 4. It can be seen that the RCS curve at this time generally shows two large bumps, which are located on the left and right sides of the tail rotor, respectively; this is because the lateral illumination area of the tail rotor is large, and many strong scattering sources will appear. The two RCS curves generally agree well, and the average RCS value determined by PO + PTD is 0.45 dBm^2^ smaller than the other, where larger RCS differences (over 1.19 dBm^2^) occur around 125.8° and 284.3°. These results show that the DSM based on PO + PTD is accurate and feasible for solving the instantaneous RCS of the tail rotor.

For the verification of dynamic RCS, the combination of QSP and FEKO is used to solve the tail rotor; that is, a finite number (here 50) of discrete states are generated in a basic passing time as shown in Figure 5, where *n*_r2_ is the rotation speed of the tail rotor. The tail rotor RCS curve determined by DSM fluctuates and shows a peak (6.71 dBm^2^ at *t* = 0.0067 s) during the current observation time. The two RCS curves are generally consistent, and the mean value of the DSM curve is 0.429 dBm^2^ smaller than the other. The results show that DSM is accurate and efficient for solving rotor dynamic RCS.

## 3. Rotor and Helicopter Models

Taking the Eurocopter Tiger as a reference object, the helicopter model in this article is established as shown in Figure 6, where *L*_fus_ is the length of the fuselage, *W*_fus_ is the width of the fuselage, *H*_fus_ is the height of the fuselage, *R*_r1_ is the radius of the main rotor, *R*_r2_ is the radius of the tail rotor, and *R*_r1h_ is the radius of the main rotor hub. In addition, Table 1 gives their specific numerical values.

The size distribution of the main rotor and tail rotor is shown in Figure 7, where *A*_t_ is the twist angle of the blade, numeric subscripts indicate different section positions, *L*_r2a_ is the length of the axis fairing of the tail rotor, and *R*_r2h_ is the hub radius of the tail rotor, as shown in Table 2.

Rotor and helicopter models use high-precision unstructured mesh technology for surface meshing as shown in Figure 8, where local mesh encryption is used to ensure the mesh quality of the rotor edges and the small parts of the helicopter. For the main rotor, the blade tip, leading edge, trailing edge, and shaft are the smaller areas. For the tail rotor, the leading, trailing, and shaft curvatures of the blades are large. For helicopter fuselage, short wing edges, weapon racks, landing gear, and vertical tail edges are smaller dimensions.

## 4. Results and Discussion

Figure 9 provides that the dynamic RCS of the tail rotor at different radar wave frequencies shows significant differences. From 1 to 5 GHz, the overall level of the RCS curve has increased significantly, while from 7 to 11 GHz, that of the RCS curve is reduced first and then increased, where the average value of the RCS curve also changes similarly, as shown in Table 3. As the incident direction of the radar wave passes through the tail-rotor disc, there are fewer surfaces forming strong scattering sources on the blade surface, which leads to a lower overall level of all RCS curves (most RCS are less than –2 dBm^2^). However, it can be found that most RCS curves will produce a large peak at 3.33 × 10^−3^ s, because at this time, the blade at the lower front position is rotated to the horizontal position, and the end face of the blade forms a strong scattering source. With the increase of radar wave frequency, the subtle surface of the blade is more likely to form a strong scattering source, which causes the overall level of the RCS curve to increase. These results indicate that DSM can describe the rotor’s dynamic RCS over a wide frequency range.

### 4.1. Effect of Rotation Speed

Figure 10 indicates that the dynamic RCS of the tail rotor at different speeds shows different cycle characteristics, and their periods are equal to their respective basic transit times. It can be found that the peak values of the six RCS curves are all 2.218 dBm^2^. As the speed of the tail rotor increases, the RCS curve is compressed against the time axis, because the base passage time becomes shorter, where the basic transit time of the RCS curve at 1200 r/min is 0.0167 s, while that of the RCS curve at 2700 r/min is 0.0074 s. The period of the RCS curve at 1200 r/min is equal to 0.0167 s, and this value is also equal to the basic transit time at this time, because when the pitch angle and roll angle are equal to 0, after the blades have rotated *A*_b,r2_, the rotation state of the tail rotor starts to repeat. These results show that the dynamic RCS period of the rotor changes with the rotation speed. When the pitch and roll angles are equal to 0, this period is equal to the basic transit time of the rotor.

Figure 11 shows that the periodicity of the dynamic rotor RCS of the main rotor decreases as its speed increases, and the RCS cycle at each speed is also equal to the current basic transit time, where *n*_r1_ is the rotation speed of the main rotor. When *n*_r1_ = 300 r/min, the RCS curve has three repetitive changes in the range of 0 to 0.15 s, and it can be inferred that this RCS period is equal to 0.05 s, while the basic transit time is also equal to 0.05 s. In addition, the peaks of these three RCS curves are equal to 21.61 dBm^2^, while the period of the RCS curve at 400 r/min is only 0.0333 s. It can be seen that although the dynamic RCS of the main rotor and the tail rotor are very different, the RCS cycle is still regular.

In general, the dynamic RCS of the main rotor and the tail rotor exhibit periodicity, and this period decreases with the increase of the rotational speed. When the pitch and roll angles are both equal to 0, the rotor’s dynamic RCS period is exactly equal to its basic transit time.

### 4.2. Effect of Observation Angles

Figure 12 manifests that azimuth angle has a significant effect on the dynamic RCS of both the main rotor and the tail rotor. For the main rotor, the fluctuation amplitudes of the RCS curves of *α* = 80° and 100° are similar—both in the range of −13.25 to 13.82 dBm^2^—while the RCS curve of *α* = 90° shows a peak value of 37.04 dBm^2^ at *t* =0 s because the end face of the main rotor with a blade tip just points toward the incident direction of the radar wave. The average dynamic RCS of the main rotor at different lateral azimuths is shown in Table 4, where the average RCS of the main rotor at 110° is only −4.23 dBm^2^, while that at 90° is as high as 17.10 dBm^2^. For the tail rotor, the RCS curve at 90° azimuth is almost straight (RCS values are all around 9.263 dBm^2^) in the current range, because the tail rotor disc is perpendicular to the radar wave at this time, while the total deflection effect of the three blades on the radar wave is almost unchanged. The shapes of the RCS curves at *α* = 80° and 100° are similar, and the peaks are equal to 18.13 dBm^2^. The average dynamic RCS level of the tail rotor in the azimuth range of 80°–120° is relatively high. The RCS mean at 120° is 4.42 dBm^2^, and that at 110° is as high as 11.78 dBm^2^. These results show that although the main rotor has a larger size, the tail rotor exhibits stronger electromagnetic scattering characteristics under the lateral azimuth.

Figure 13 investigates that the elevation angle has a significant effect on the dynamic RCS of the main rotor and tail rotor. For the main rotor, there is only one peak in the RCS curve at *β* = 0°, with a size of 10.1 dBm^2^ at 0.0033 s, while three peaks appear in the RCS curve at *β* = 5°, which are 11.17 dBm^2^ at 0.0033 s, 11.79 dBm^2^ at 0.0153 s, and 10.93 dBm^2^ at 0.0261 s, respectively. The RCS curve at *β* = 10° has only one peak, but its overall level is further reduced, where the dynamic RCS mean of the main rotor is shown in Table 5. When *β* increases from −10° to 10°, the average dynamic RCS of the main rotor increases first and then decreases, because the strong scattering source on the surface of the hub has changed significantly at this time, when *β* is equal to 0 °, the hub is in the position that is most detrimental to deflecting radar. For the tail rotor, the effect of elevation angle on the RCS curve mainly includes the shape, peak, and minimum value. When *β* increases from −10° to 10°, the mean dynamic RCS of the tail rotor decreases first and then increases, while these RCS averages are generally large because the angle between the tail rotor disk and the radar wave is large. These results indicate that the elevation angle has a great impact on the dynamic RCS of the rotor, where the average RCS of the main rotor changes greatly, but the overall level is low, while the average RCS of the tail rotor has less change, but the overall level is high.

### 4.3. Effect of Pitching Angle

Figure 14 shows that the pitch angle has a significant effect on the dynamic electromagnetic scattering on the surface of the helicopter. For Figure 14a, the main rotor rotates 8.1° and the tail rotor rotates 24.3°, where crimson is more distributed in the nose, the front of the cockpit, the top of the fuselage, the front of the hub of the main rotor, the leading edge of the short wing, the front of the external weapon, the landing gear, the leading edge of the horizontal tail, and the front of the vertical tail. When the time advances to 0.0087 s, the rear part of the fuselage’s tail beam turns green, and the blade surfaces of the main rotor and tail rotor also change dramatically. This is because the pitch angle can change the angle between the rotor disk and the radar wave. Adding the rotation of the rotor itself results in a dynamic change in its electromagnetic scattering characteristics. These results show that DSM can well describe the dynamic effects of the main rotor and tail rotor on the electromagnetic scattering characteristics of helicopters.

Figure 15 reveals that pitch angle will have a greater impact on the dynamic RCS of the helicopter. For Figure 15a, most of the RCS values of the curve at *θ* = 0° are above 20 dBm^2^, and the average value reaches 48.84 dBm^2^ as shown in Table 6, while the overall level of the RCS curve at *θ* = 10° is much lower than that of the RCS curve at 0°, where the mean value of the RCS curve at *θ* = 10° is 40.95 dBm^2^. For Figure 15b, the helicopter’s RCS curve still shows dynamic changes at different time points, although the pitching angle is equal to 10°, because of the electromagnetic scattering changes caused by the rotation of the main rotor and tail rotor. The overall trend of the RCS curve is similar at *t* = 0.0177 s and 0.0126 s, and they both produce peaks at 90° and 270° because large area flat plates on both sides of the fuselage, external weapons, tall vertical tail, and fairing of the intake and exhaust system all use mediocre design, which results in their weak ability to deflect lateral radar waves.

In general, the pitching angle change has a great effect on the helicopter RCS curve and average. Although the RCS contribution of the fuselage is large, it still cannot cover the dynamic changes brought about by the rotor.

### 4.4. Effect of Rolling Angle

Figure 16 presents that the rolling angle has a significant effect on the dynamic electromagnetic scattering characteristics of the helicopter surface. For Figure 16a, the main rotor rotates 58° and the tail rotor rotates 174°. At this time, a large area of deep red appeared on the nose and the front of the cockpit, the front of the main rotor hub shows a transition from dark red to red, the surface of some main rotor blades has more red, the front of the landing gear and short wings are also dark red concentrated, tail rotor hub and vertical tail show red. For Figure 16b, the main rotor rotates 168° and the tail rotor rotates 504°, where the red area on the main rotor surface is significantly reduced and the dark blue area is increased, while the large red and yellow areas on the fuselage surface have also become lighter, including the nose, cockpit, fuselage side, tail beam, horizontal tail, and vertical tail. The increase in roll angle increases the tilt angle between the side of the fuselage and the radar wave, making it easier for the incident wave to be scattered to the upper and rear areas. This change is also applicable to the interpretation of the special changes in the scattering of the vertical tail lighting. However, the overall effect of deflecting radar waves in its lighting area has not been improved because the main rotor’s hub is designed as a rotating body (combination of ellipsoid and cylinder). These results indicate that it is feasible for DSM to describe the effect of attitude angle changes on helicopter dynamic RCS.

Figure 17 provides that the rolling angle has a significant effect on both the instantaneous and dynamic RCS of the helicopter. For Figure 17a, the RCS curves at *γ* = 5° and 10° are generally similar, but there are large differences in the local fluctuations and peaks, where the maximum value of the RCS curve at *γ* = 5° is 39.55 dBm^2^ at 265.5°, while that of the RCS curve at *γ* = 15° is 50.92 dBm^2^ at 228°. In addition, the difference between the two RCS curves in the azimuth ranges of 71–114.5° and 224.3–291° mostly exceeds 10 dBm^2^, because when the roll angle is changed, the electromagnetic scattering characteristics of the fuselage have changed greatly, and the main rotor and tail rotor also make a certain contribution to the lateral RCS. However, the influence of the fuselage is still major, so these two RCS curves are similar. For Figure 17b, the RCS curves at *γ* = 10° and 15° both show dynamic characteristics, and the average RCS value of the former is 4.54 dBm^2^ higher than the latter. The RCS curve at *γ* = 10° shows two large peaks, 16.72 dBm^2^ at 0.0042 s and 18.71 dBm^2^ at 0.0177 s, while the RCS curve at *γ* = 15° shows three large peaks, 18.26 dBm^2^ at 0.0054 s, 18.22 dBm^2^ at 0.0186 s, and 28.8 dBm^2^ at 0.0234 s. The reason for these changes is mainly that the rotation of the tilted main rotor brings dynamic electromagnetic scattering characteristics to the helicopter, plus the dynamic scattering contribution of the tail rotor. Table 7 provides the average RCS of the helicopter at different roll angles, showing that the average RCS of the helicopter reaches 40.81 dBm^2^ at *γ* = 0°, while the average RCS of the helicopter is only 10.41 dBm^2^ at a roll angle of −20°. These results indicate that the roll angle has a greater impact on the RCS of the helicopter, but it still cannot cover the dynamic changes brought about by the rotor components.

## 5. Conclusions

Based on the established dynamic electromagnetic scattering method, the RCS of the rotors and the helicopter were thoroughly studied and discussed. By studying the static and dynamic RCS of the helicopter, the following conclusions can be drawn:(1)The RCS of the main rotor and tail rotor is indeed dynamic and periodic, and their periodic characteristics are related to the rotation speed and attitude angle;(2)Increasing the rotor speed can significantly reduce its dynamic RCS period, but it cannot change its fluctuation amplitude and peak value;(3)Azimuth and elevation angle are also important factors affecting helicopter RCS. When the radar wave illuminates the tail rotor disc vertically, the dynamic RCS characteristics of the tail rotor are negligible.(4)The pitching and rolling angles have a large impact on the static and dynamic RCS of the helicopter, but they still cannot cover the dynamic electromagnetic scattering characteristics brought by the main rotor and tail rotor.

## Figures and Tables

**Figure 1 sensors-20-02097-f001:**
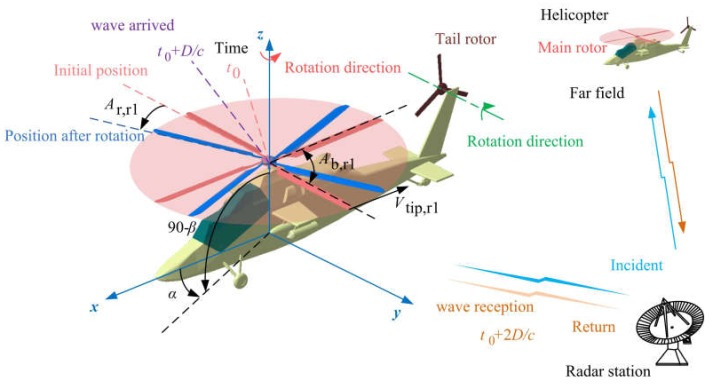
Schematic diagram of the effect of rotor rotation on the helicopter radar cross-section.

**Figure 2 sensors-20-02097-f002:**
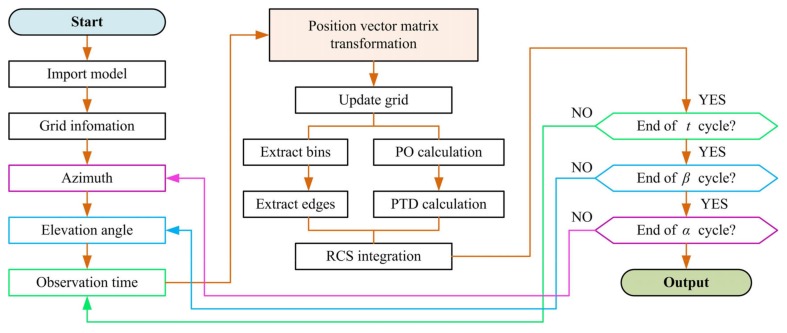
Method flow chart.

**Figure 3 sensors-20-02097-f003:**
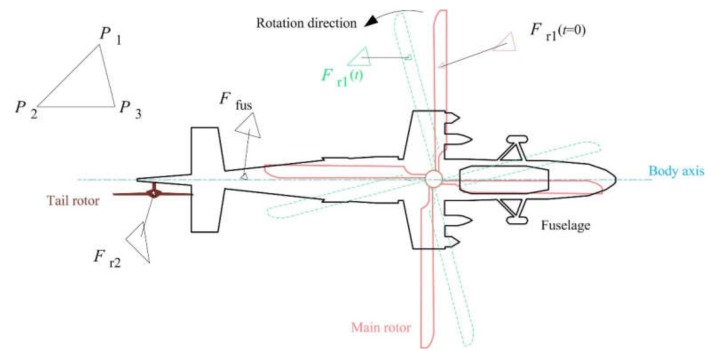
Schematic of the illuminated area change of the helicopter surface when the rotor is rotating.

**Figure 4 sensors-20-02097-f004:**
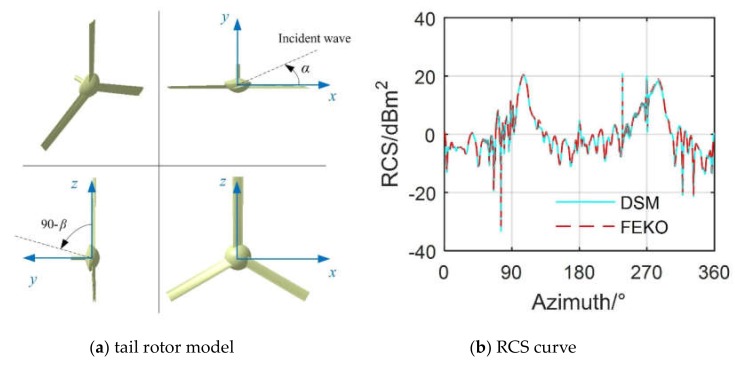
Validation of the radar cross-section (RCS) algorithm on the tail rotor, *f*_R_ = 5 GHz, *α* = 0°~360°, *β* = 0°, *t* = 0 s.

**Figure 5 sensors-20-02097-f005:**
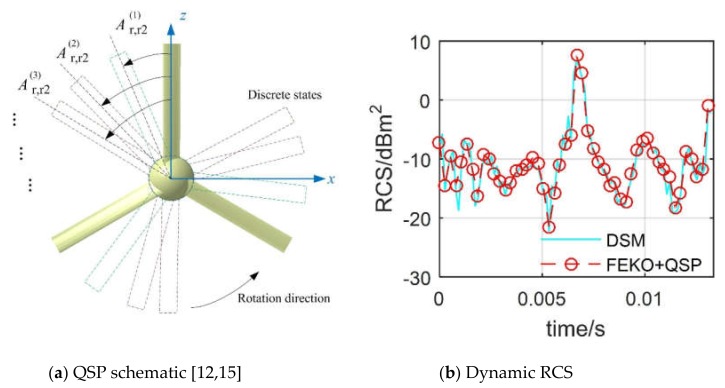
Validation of dynamic RCS on the tail rotor, *f*_R_ = 5 GHz, *α* = 60°, *β* = 0°, *n*_r2_ = 1500 r/min.

**Figure 6 sensors-20-02097-f006:**
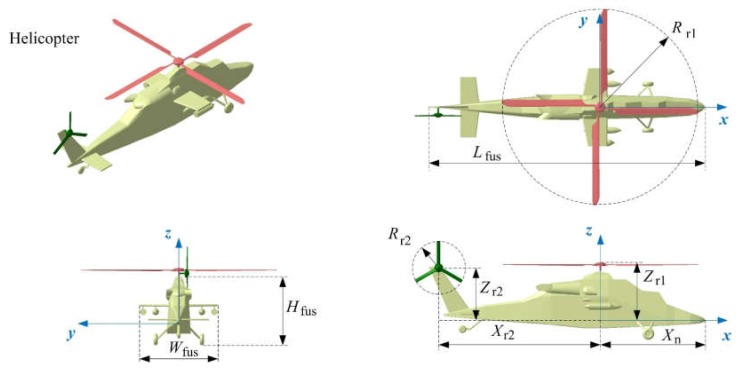
Helicopter model and its main dimensions.

**Figure 7 sensors-20-02097-f007:**
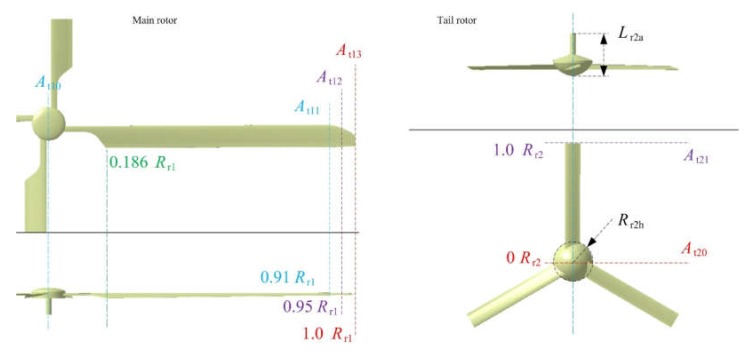
Main rotor and tail rotor size distribution.

**Figure 8 sensors-20-02097-f008:**
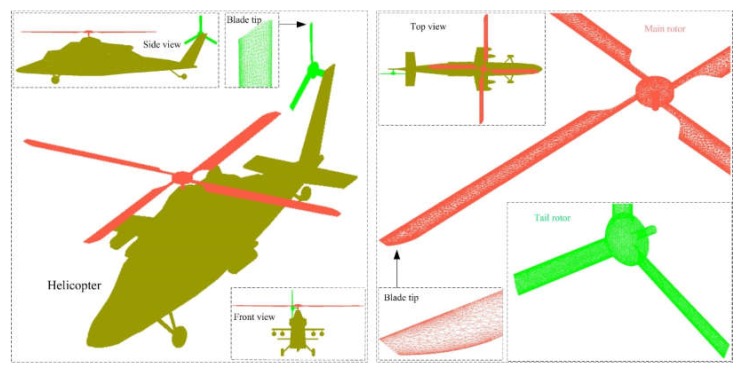
Meshing of helicopter and rotor model surfaces.

**Figure 9 sensors-20-02097-f009:**
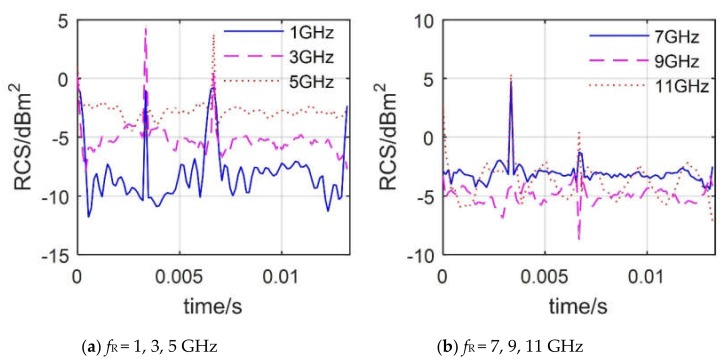
Dynamic RCS of the tail rotor at different *f*_R_ values, *α* = 0°, *β* = 0°, *n*_r2_ = 1500 r/min.

**Figure 10 sensors-20-02097-f010:**
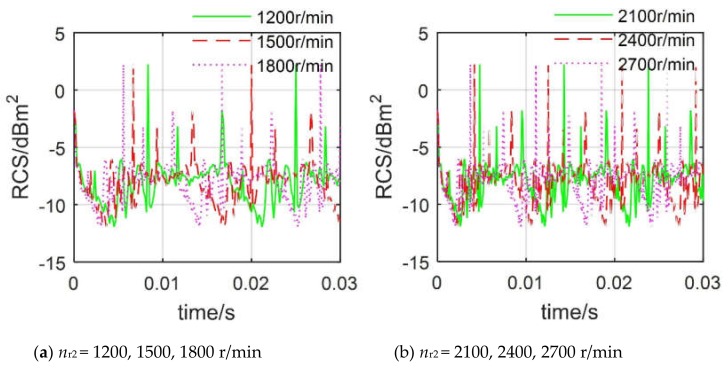
Dynamic RCS of the tail rotor at different *n*_r2_, *f*_R_ = 5 GHz, *α* = 330°, *β* = 0°, *γ* = *θ* = 0°.

**Figure 11 sensors-20-02097-f011:**
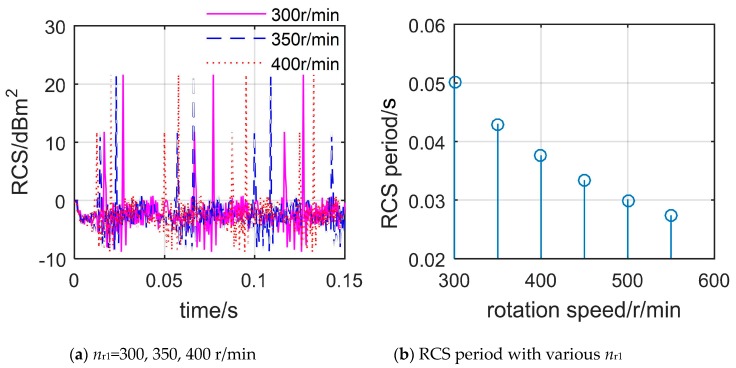
Dynamic RCS of the tail rotor at different *n*_r1_, *f*_R_ = 5 GHz, *α* = 330°, *β* = 0°, *γ* = *θ* = 0°.

**Figure 12 sensors-20-02097-f012:**
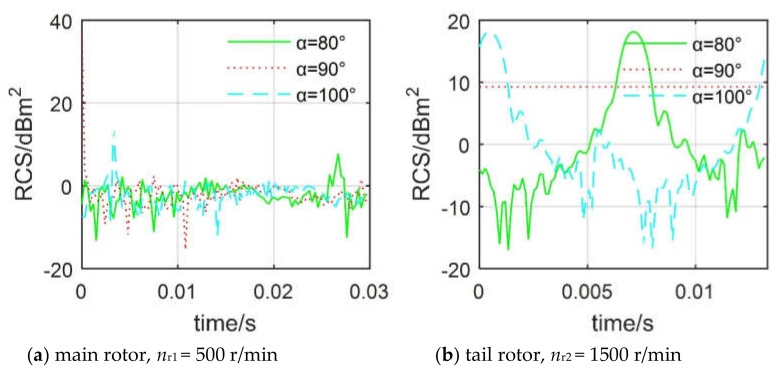
Effect of azimuth on rotor dynamic RCS, *f*_R_ =5 GHz, *β*=0°, *γ* = *θ* = 0°.

**Figure 13 sensors-20-02097-f013:**
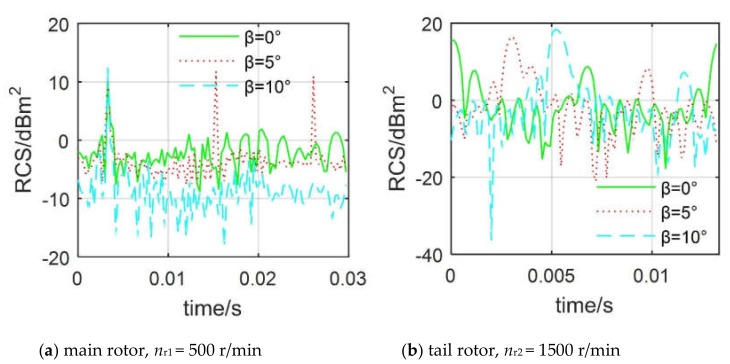
Effect of elevation angle on rotor dynamic RCS, *f*_R_ = 5 GHz, *α* = 280°, *γ* = *θ* = 0°.

**Figure 14 sensors-20-02097-f014:**
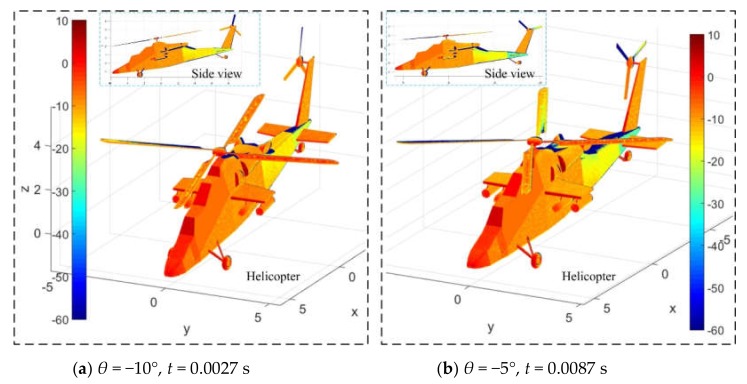
Effect of pitching angle on surface electromagnetic scattering of the helicopter, *n*_r1_ = 500 r/min, *n*_r2_ = 1500 r/min, *f*_R_ = 5 GHz, *α* = 25°, *γ* = 0°.

**Figure 15 sensors-20-02097-f015:**
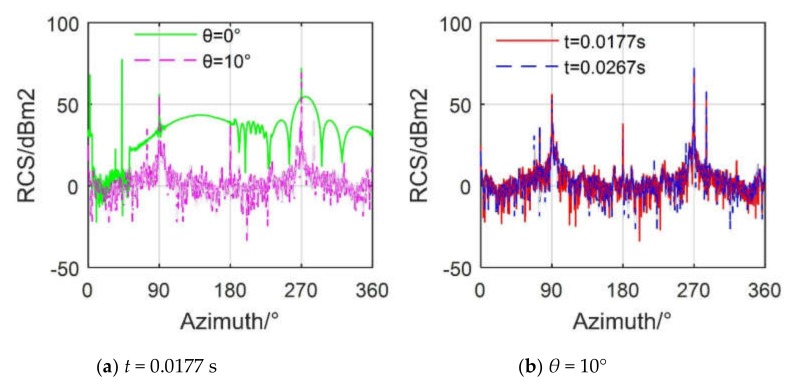
Effect of pitching angle on helicopter RCS, *n*_r1_ = 500 r/min, *n*_r2_ = 1500 r/min, *f*_R_ = 5 GHz, *γ* = 0°.

**Figure 16 sensors-20-02097-f016:**
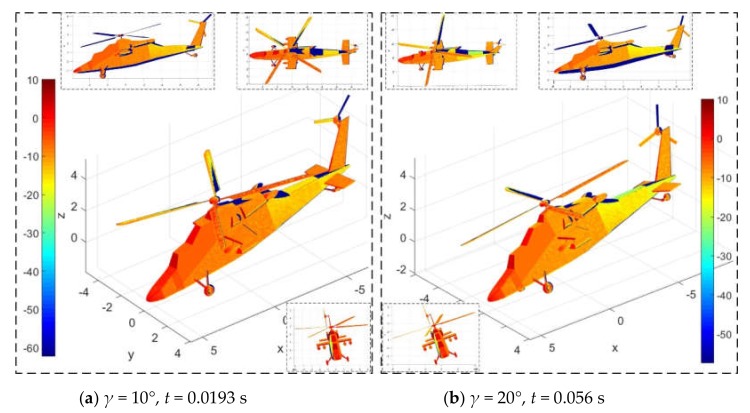
Effect of rolling angle on helicopter surface scattering, *n*_r1_ = 500 r/min, *n*_r2_ = 1500 r/min, *f*_R_ =5 GHz, *α* = 30°, *β* = 0°, *θ* = −10°.

**Figure 17 sensors-20-02097-f017:**
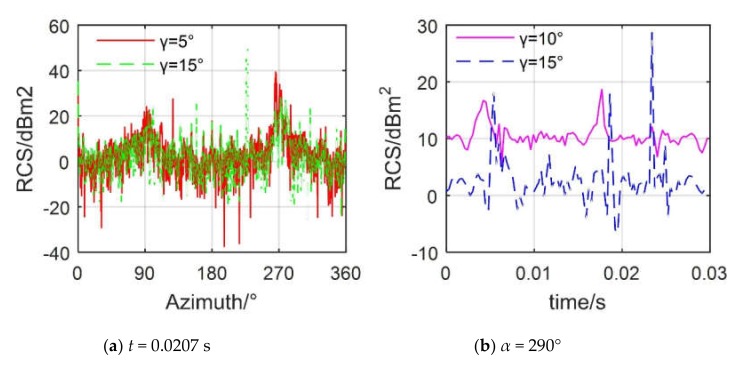
Effect of rolling angle on helicopter RCS, *n*_r1_ = 500 r/min, *n*_r2_ = 1500 r/min, *f*_R_ = 5 GHz, *β* = 0°, *θ* = −10°.

**Table 1 sensors-20-02097-t001:** The main dimensions of the helicopter model.

Parameter	*L*_fus_/m	*W*_fus_/m	*H*_fus_/m	*X*_n_/m	*Z*_r1_/m
Value	15.546	4.4	3.84	5.9	3.1
Parameter	*R*_r1_/m	*R*_r2_/m	*R*_r1h_/m	*X*_r2_/m	*Z*_r2_/m
Value	5.5	1.5	0.3	9.1	2.8

**Table 2 sensors-20-02097-t002:** The main dimensions of the helicopter model.

Main Rotor	*A*_t10_/°	*A*_t11_/°	*A*_t12_/°	*A*_t13_/°	Airfoil
Value	13	5	4	3	HD53
Tail rotor	*A*_t20_/°	*A*_t21_/°	*R*_r2h_/m	*L*_r2a_/m	Airfoil
Value	20	10	0.25	0.54	Dormoy

**Table 3 sensors-20-02097-t003:** RCS mean of the tail rotor at different *f*_R_ values, *α* = 0°, *β* = 0°, *n*_r2_ = 1500 r/min.

***f*_R_/GHz**	1	3	5	7	9	11
**Mean RCS/dBm^2^**	−7.20	−4.74	−2.65	−2.89	−4.87	−3.24

**Table 4 sensors-20-02097-t004:** Effect of azimuth on rotor dynamic RCS mean, *f*_R_ = 5 GHz, *β* = 0°, *γ* = *θ* = 0°.

**Azimuth/°**	80	90	100	110	120
**Main Rotor/dBm^2^**	−1.28	17.10	−0.42	−4.23	−1.21
**Tail Rotor/dBm^2^**	7.78	9.26	7.78	11.78	4.42

**Table 5 sensors-20-02097-t005:** Effect of elevation angle on rotor dynamic RCS mean, *f*_R_ = 5 GHz, *α* = 280°, *γ* = *θ* = 0°.

***β*/°**	−10	−5	0	5	10
**Main Rotor/dBm^2^**	−9.25	−4.96	−1.20	−0.85	−4.64
**Tail Rotor/dBm^2^**	7.68	4.66	4.37	4.68	6.45

**Table 6 sensors-20-02097-t006:** Effect of pitching angle on helicopter RCS mean, *n*_r1_ = 500 r/min, *n*_r2_ = 1500 r/min, *f*_R_ = 5 GHz, *α* = 0~360°, *γ* = *β* = 0°, *t* = 0.0177 s.

***θ*/°**	−10	−5	0	5	10
**RCS Mean/dBm^2^**	40.93	40.80	48.84	40.81	40.95

**Table 7 sensors-20-02097-t007:** Effect of rolling angle on helicopter RCS mean, *n*_r1_ = 500 r/min, *n*_r2_ = 1500 r/min, *f*_R_ =5 GHz, *α* = 0~360°, *β* = 0°, *θ* = −10°, *t* = 0.0207 s.

***γ*/°**	−20	−15	−10	0	10	15	20
**RCS Mean/dBm^2^**	10.41	21.13	14.66	40.81	11.51	21.46	11.25

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
