# Peer review of "Influence of Rotor Dynamic Scattering on Helicopter Radar Cross-Section"

_sensors, 2020, doi:10.3390/s20072097_

Round 1

Reviewer 1 Report

The paper is quite interesting and well written. It offers an adequate theoretical background and well presented results.

However, some more details concerning the simulation section would be appreciated. For example, a point which is not so well clarified is the software used to perform the RCS calculations with the proposed Dynamic Scattering Method. I understand that you compare your results with the FEKO software. However, which computer program have you used to obtain your results? Was it MATLAB, C++ code or something else? What was the size of the mesh used for the fuselage and the rotors? Furthermore, it would be interesting to provide some information concerning the computational requirements, e.g., X hours on an Y class computer for a specific run, both for your proposed method, as well as for FEKO.

Some minor suggestions follow:
line 25: Sikorsky X-2 → Sikorsky X2, Euro copter → Eurocopter
line 29: "which makes important components considered radar stealth from the beginning" requires rephrasing, such as "requiring the consideration of radar stealth for important components from the beginning"
line 32: unique flight → unique flight behaviour
line 46: equivalent current method → an equivalent current method
line 81: scattering → scattered
line 87-88: "M (r) is to the equivalent current of the panel of the illuminated area, J(r) is the equivalent magnetic flux of the panel of the illuminated area" you may consider the following "M (r) is the equivalent magnetic surface current of the panel of the illuminated area, J(r) is the equivalent electric surface current of the panel of the illuminated area"
line 100: y(P)i → y(Pi)
line 107: y(P)i → y(Pi), Fr1 → Fr2
line 124: there are → we have
line 126: "and Ipis to the feature quantity integral of the p-th triangle face element" requires rephrasing
line 129: a reference offering some details on "the Gardon method" would be appreciated
line 140: (3,5) ???
line 175: (12,15) ???
line 217: Figure 8 → Figure 9
line 351: Effect of elevation angle → Effect of rolling angle
line 361-3: Should be rephrased, possible by putting a full stop after "RCS".
Finally, the nomenclature of the variable tb, i.e. "base transit time of the rotor blade[s]", is not so clear.

Author Response

Responds to the reviewer’s comments:

Reviewer #1:

  1. However, some more details concerning the simulation section would be appreciated. For example, a point which is not so well clarified is the software used to perform the RCS calculations with the proposed Dynamic Scattering Method. I understand that you compare your results with the FEKO software. However, which computer program have you used to obtain your results? Was it MATLAB, C++ code or something else? What was the size of the mesh used for the fuselage and the rotors? Furthermore, it would be interesting to provide some information concerning the computational requirements, e.g., X hours on an Y class computer for a specific run, both for your proposed method, as well as for FEKO.

Response:

We have no software to simulate the dynamic RCS of a helicopter with multiple rotors, and it is difficult for FEKO to give the large number of RCS-t results required. Therefore, the author developed a dynamic calculation program based on Matlab, which can not only solve the static electromagnetic scattering characteristics, but also simulate the dynamic RCS, and promote it to high-speed helicopters and coaxial helicopters. The difficulty of the problem is that the main rotor and tail rotor shafts are different and the rotation speeds are not synchronized. There is a high-speed relative rotation movement between the rotor and the fuselage. This results in the conventional methods (QSP and relativistic methods) and algorithms (MOM and PO), etc. cannot directly and efficiently solve the dynamic RCS.

We use high-precision unstructured grid technology to discretize the rotor and helicopter models, using a 2-4 mm encryption size on the leading and trailing edges of each blade, and 5 mm encryption on other surfaces with large curvature changes. When performing FEKO simulation, Hypermesh is used for grid division and the grid size is uniformly 6 mm, parallel computing with server. It takes more than 6 hours to solve a RCS-α curve using FEKO where â–³α = 0.25°. Solving RCS-t is calculated separately at each time scale.

  1. line 25: Sikorsky X-2 → Sikorsky X2, Euro copter → Eurocopter.

Response:

Considering the Reviewer’s suggestion, we made these changes as shown in the manuscript.

  1. line 29: "which makes important components considered radar stealth from the beginning" requires rephrasing, such as "requiring the consideration of radar stealth for important components from the beginning".

Response:

We have modified the expression of this sentence.

  1. line 32: unique flight → unique flight behavior.

Response:

This wording has been added as shown in the context.

  1. line 46: equivalent current method → an equivalent current method.

Response:

We perfected this phrase.

  1. line 81: scattering → scattered.

Response:

It is really true as Reviewer suggested that.

  1. line 87-88: "M (r) is to the equivalent current of the panel of the illuminated area, J(r) is the equivalent magnetic flux of the panel of the illuminated area" you may consider the following "M (r) is the equivalent magnetic surface current of the panel of the illuminated area, J(r) is the equivalent electric surface current of the panel of the illuminated area".

Response:

The modification has been completed as shown in red font in section 2.1.

  1. line 100: y(P)i → y(Pi).

Response:

It is really true as Reviewer suggested that this subscript is misplaced. The modification is completed as shown in the text.

  1. line 107: y(P)i → y(Pi), Fr1 → Fr2.

Response:

Both subscripts have been modified as shown in the manuscript.

  1. line 124: there are → we have.

Response:

This short sentence has been revised as suggested.

  1. line 126: "and Ipis to the feature quantity integral of the p-th triangle face element" requires rephrasing.

Response:

It is really true as Reviewer suggested that this sentence needs to be modified. The original sentence was changed to ‘and Ip is the feature integration of the p-th triangle face element.’

  1. line 129: a reference offering some details on "the Gardon method" would be appreciated.

Response:

The literature (reference [27]) has been supplemented as shown at the end of the manuscript and cited in the text.

  1. line 140: (3,5) ???.

Response:

The reference symbol has been modified.

  1. line 175: (12,15) ???

Response:

The relevant modification is completed.

  1. line 217: Figure 8 → Figure 9.

Response:

It is really true as Reviewer suggested that the picture numbers are not continuous. The modification has been completed.

  1. line 351: Effect of elevation angle → Effect of rolling angle.

Response:

We made changes and checked other descriptions as shown in section 4.3 and 4.4.

  1. line 361-3: Should be rephrased, possible by putting a full stop after "RCS".

Response:

It is really true as Reviewer suggested that.

  1. Finally, the nomenclature of the variable tb, i.e. "base transit time of the rotor blade[s]", is not so clear.

Response:

It is really true as Reviewer suggested that tb needs more explanation. As explained after formula (28) in the text, tb is equal to the time it takes for the blade to rotate through the angle between two adjacent blades.

Special thanks to the editors and reviewers for their comments and suggestions. The reviewer's meticulous and rigorous and realistic style is admirable and worth learning. This reviewer's suggestion makes this article more complete and rich. Thanks again.

Reviewer 2 Report

The authors provide a deep study of the RCS of helicopters considering the rotation of rotor’s blades when evaluating the electromagnetic scattering interactions and the variability in the observed target.

In the Reviewer’s opinion, the provided analyses are complete and very interesting, but the paper should be improved in some respects.

The novel contribution of the paper should be better emphasized in Introduction.

The English usage needs some improvements. Please, make a complete proofread of the manuscript.

I suggest a synthetic representation of the overall model, like a block scheme, or some useful graphical descriptor.

A Notation section would help the reader in better understanding the equations.

Author Response

Responds to the reviewer’s comments:

Reviewer #1:

  1. The novel contribution of the paper should be better emphasized in Introduction.

Response:

Considering the Reviewer’s suggestion, we elaborated on the innovations in this article in the penultimate paragraph of the introduction.

  1. The English usage needs some improvements. Please, make a complete proofread of the manuscript.

Response:

As Reviewer suggested that the language of this article needs improvement. We have carried out a detailed inspection and revision of the full text as shown in the manuscript.

  1. I suggest a synthetic representation of the overall model, like a block scheme, or some useful graphical descriptor.

Response:

It is really true as Reviewer suggested that the comprehensive expression is necessary. We added a flowchart of the entire method so that readers can better grasp this process as shown in Section 2.1.

  1. A Notation section would help the reader in better understanding the equations.

Response:

It is really true as Reviewer suggested that these formulas need clearer notes. We have further refined the interpretation of the formula, as shown in Section 2.

Special thanks to the editors and reviewers for their comments and suggestions. These changes make this article more beautiful and easy to understand, thanks again.
